# Increased elastase sensitivity and decreased intramolecular interactions in the more transmissible 501Y.V1 and 501Y.V2 SARS-CoV-2 variants' spike protein–an *in silico* analysis

Suman Pokhrel[☯], Benjamin R. Kraemer[ID][☯], Lucia Lee[☯], Kate Samardzic, Daria Mochly-Rosen[ID]*

Department of Chemical and Systems Biology, Stanford University School of Medicine, Stanford, CA, United States of America

☯ These authors contributed equally to this work.
* mochly@stanford.edu

**Data Availability Statement:** All relevant data are within the paper and its Supporting Information files.

## Abstract

Two SARS-CoV-2 variants of concern showing increased transmissibility relative to the Wuhan virus have recently been identified. Although neither variant appears to cause more severe illness nor increased risk of death, the faster spread of the virus is a major threat. Using computational tools, we found that the new SARS-CoV-2 variants may acquire an increased transmissibility by increasing the propensity of its spike protein to expose the receptor binding domain via proteolysis, perhaps by neutrophil elastase and/or via reduced intramolecular interactions that contribute to the stability of the closed conformation of spike protein. This information leads to the identification of potential treatments to avert the imminent threat of these more transmittable SARS-CoV-2 variants.

## Introduction

Severe acute respiratory syndrome coronavirus 2 (SARS-CoV-2), the novel coronavirus that has resulted in the coronavirus disease 2019 (COVID-19) pandemic, continues to mutate while spreading throughout the world. A critical protein on the surface of the virus is the spike protein, that mediates the entry of the virus into the host cells [1]. When a mutation or set of mutations provide an advantage over the previous variants, the new variant becomes the dominant variant to spread. This is what occurred when the aspartate 614 in the spike protein of the initial (Wuhan) SARS-CoV-2 virus, mutated to a glycine (Asp614Gly); within a month, the new variant became the most dominant in Europe [2] and now globally [3]. The Asp614Gly variant is associated with higher viral loads, but not with a more severe COVID-19 symptoms in infected individuals [4]. Importantly, the Asp614Gly mutation does not affect the ability of antibodies to neutralize that variant [5]. Nevertheless, as the virus continues to mutate at a very high rate, [for example, each amino acid in the 1273 amino acid of the viral spike protein has mutated on average almost 3 times per position since the protein was first sequenced, about a year ago [6], the need to continue active surveillance for the emergence of new variants

**Funding:** Supported in part by the 2020 COVID-19 Response: Drug and Vaccine Prototyping Grant from the Innovative Medicines Accelerator, Stanford University to D. M.-R. and by the SPARK at Stanford community. The funders had no role in study design, data collection and analysis, decision to publish, or preparation of the manuscript.

**Competing interests:** The authors have declared that no competing interests exist.

**Abbreviations:** hACE2, Human Angiotensin converting enzyme 2; MOE, Molecular Operating Environment; PDB, Protein Data Bank; RBD, Receptor binding domain.

and examination of means to slow down the spread of the new variants is of utmost importance.

## The SARS-CoV-2 United Kingdom variant, 501Y.V1, and the South African variant, 501Y.V2

A novel SARS-CoV-2 variant, known as B.1.1.7 or 20B/501Y.V1, emerged in the United Kingdom at the end of September 2020 and became the dominant variant within a month [7]. This variant increases viral transmissibility between 40–70% (the range reflects increased transmission over whatever variants were circulating at the time and place of the report [8, 9]. As of January 16 2021, 501Y.V1 has been reported in more than 50 countries [10]. However, this new variant does not appear to increase the severity of COVID-19 [11]. The variant has concomitant 3 deletions and 10 amino acid changes in the 1273 amino acid spike protein (Table 1), compared with the initial SARS-CoV-2 index virus identified in Wuhan, China; only one (Asn501Tyr or N501Y) is in the human angiotensin-converting enzyme 2 (hACE2) receptor-binding domain (RBD; amino acids 331–524) [12]. Since this Asn501Tyr mutation in the spike's RBD was observed alone as early as April 2020 in Brazil and later in Australia without reports of increased transmissibility [13], it is unlikely that this single substitution is sufficient to explain the new phenotype of B.1.1.7 variant.

De Oliveira and collaborators identified another more transmittable and dominant variant, termed 501Y.V2 (aka 20H/501Y.V2 or B.1.351) South African variant [14]. Identified first in the second week of October, this variant became dominant in South Africa within a month. Six fixed substitutions in all the South African variants were identified (Table 2): Asn501Tyr (identical to the 501Y.V1 UK variant [12]) and Asp614Gly (identical to the European dominant variant, identified between March and April of 2020 [2]), the Asp80Ala, Lys417Asn, Glu484Lys and Ala701Val. It appears that the combination of these substitutions results in increased infectivity without increasing COVID-19 severity [14].

Here we focused on mutations in the SARS-CoV-2 viral surface spike protein to examine how the new variant became more infective. We used computational methods, including Molecular Operating Environment (MOE) analysis [15] and software to predict the outcome

**Table 1. Predicted biological impact of mutations in the SARS-CoV-2 501Y.V1 spike protein.**

| SARS-CoV-2 501Y.V1 | | |
|---|---|---|
| **Variation** | **PROVEAN** | **SIFT** |
| His69 Deletion | N/A | N/A |
| Val70 Deletion | N/A | N/A |
| Tyr145 Deletion | N/A | N/A |
| Asn501Tyr* | Neutral | Tolerated |
| Ala570Asp | Neutral | Tolerated |
| Asp614Gly* | Neutral | Tolerated |
| Pro681His | Neutral | Tolerated |
| Thr716Ile | Deleterious | Deleterious |
| Ser982Ala | Neutral | Tolerated |
| Asp1118His | Neutral | Tolerated |

Predicted effects of amino acid substitutions common in SARS-CoV-2 501Y.V1 using Protein Variation Effect Analyzer (PROVEAN) [46] and SIFT [47, 48]. Variants predicted to have a deleterious impact on spike protein are shaded red.

*Mutation common to both variants. N/A = not applicable as software does not make predictions about deletions.

**Table 2. Predicted biological impact of mutations in the SARS-CoV-2 501Y.V2 spike protein.**

| SARS-CoV-2 501Y.V2 | | |
|---|---|---|
| **Variation** | **PROVEAN** | **SIFT** |
| Leu18Phe | Neutral | Tolerated |
| Asp80Ala | Neutral | Tolerated |
| Asp215Gly | Neutral | Tolerated |
| Leu242 Deletion | N/A | N/A |
| Leu242-244 Deletion | N/A | N/A |
| Arg246Ile | Neutral | Deleterious |
| Lys417Asn | Neutral | Tolerated |
| Glu484Lys | Neutral | Tolerated |
| Asn501Tyr* | Neutral | Tolerated |
| Asp614Gly* | Neutral | Tolerated |
| Ala701Val | Neutral | Tolerated |

Predicted effects of amino acid substitutions common in SARS-CoV-2 501Y.V2 using Protein Variation Effect Analyzer (PROVEAN) [46] and SIFT [47, 48]. Variants predicted to have a deleterious impact on spike protein are shaded red.

*Mutation common to both variants. N/A = not applicable as software does not make predictions about deletions.

of substitutions on the protein structure, to examine the features acquired by the new variants that enable them to increase the rate of infection and spreading without increasing the severity of COVID-19, the pathology resulting from the infection. The goal of this study is to determine whether the acquisition of the more infective phenotype (resulting in a more efficient and rapid viral transmission) exposes new vulnerabilities to target these variants with current drugs.

## Results

### Potential acquired features of the SARS-CoV-2 501Y.V1 and SARS-CoV-2 501Y.V2 variants

The SARS-CoV-2 spike protein is a trimer protein on the surface of the virus. This protein mediates the binding to the human ACE2 (hACE2) receptor on the nasal mucosa [16]. Each monomer in the trimeric spike protein is found in two conformations: a closed conformation, in which the RBD is folded in and is inaccessible to hACE2, and the open conformation–induced by proteolysis of the spike protein, in which the RBD becomes exposed [17]. When examining the location of the various mutations associated with increased transmissibility of 501Y.V1 and 501Y.V2, we found that these mutations are not located in a particular domain in the spike protein, nor are they all exposed in the closed or the open conformation (Fig 1A–1C).

As neither variant with increased transmissibility causes more severe COVID-19 disease [3, 11], and the mutations do not cluster to a particular site in the spike protein, we next examined the possibility that the acquired features in the new variants may lie in increased susceptibility of the spike protein to activation due to introduction of additional protease-activating sites. We hypothesized that the proteases responsible for this gain-of-function should be relatively unique to the initial site of SARS-CoV-2 entry, the nasal mucosa, as viral effects on internal organs do not seem to become more severe. At least seven ubiquitously expressed proteases have been suggested to act on the spike protein: TMPRSS2, furin, PC1, matriptase, trypsin, cathepsin L and/or cathepsin B [18, 19].

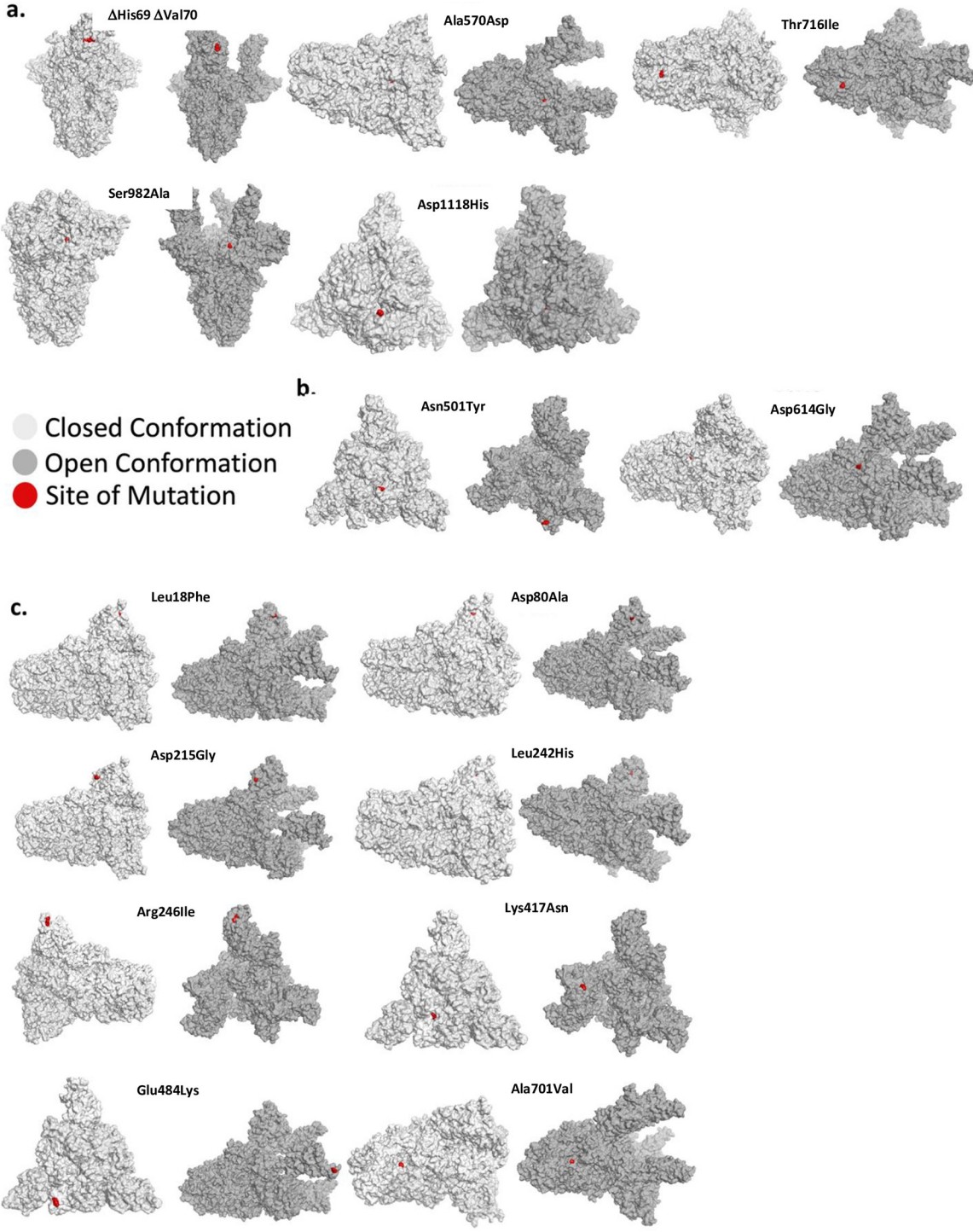

**Fig 1. Mutant position in the open and closed conformation of the spike protein. (a)** The position of various mutations of the Y501. V1, **(b)** The Asn501Tyr and Asp614Gly mutations, which occur in both the Y501.V1 and Y501.V2 variants, and **(c)** Mutations unique for the 501Y.V2. Each mutation is shown in red in one of the monomers in the 3D structure of the spike protein trimer. Shown is the closed conformation (light gray) and the open (dark gray) conformation.

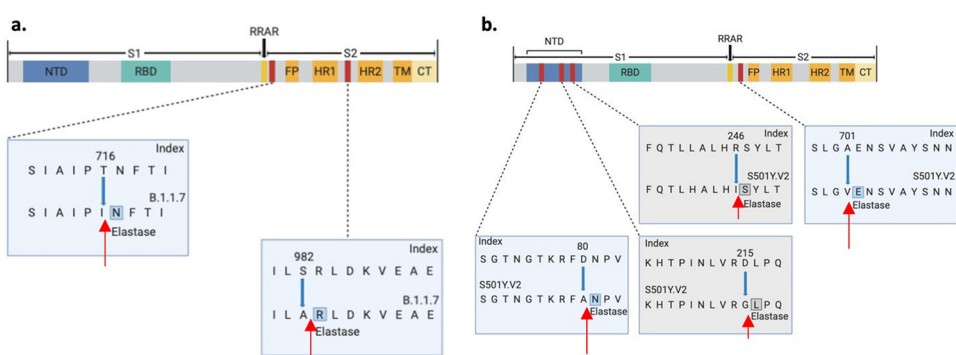

**Fig 2. Potential elastase sites in the spike protein of the new SARS-CoV-2 variants.** Potential new elastase sites in (**a**) the more infective UK variant, 501Y.V1 and (**b**) the more infective South Africa variant, 501Y.V2. Images generated using BioRender. Blue boxes indicate fixed mutations and gray boxes–non-fixed in all the sequenced 501Y. V2 [14]. Red arrows indicate the predicted cleavage site of neutrophil elastase.

In contrast to the above proteases, the neutrophil elastase is enriched in the nose of humans compared to other tissues [20] and neutrophils, a source for this elastase, are recruited to the nose in SARS-CoV-2 infection [21]. Indeed, the now worldwide dominant Asp614Gly variant, introduced a new elastase proteolysis site in the spike protein [2, 22]. Human neutrophil elastase prefers aliphatic amino acids (i.e. valine, alanine, isoleucine, and to a much lesser extent threonine) at the P1 site of cleavage [23–25]. Using ExPASy [26], we identified Thr716Ile and Ser982Ala in 501Y.V1 as potential new elastase cleavage sites (Fig 2A). Although human neutrophil elastase may cleave the protein with Thr at the P1 position, its propensity to cleave with Ile at the P1 site is much greater and the preference of the protease may also depend on the residues surrounding the cleavage site [23, 24]. Thr716 is on the surface of the spike protein whereas Ser982 is barely exposed in the open (active) conformation and is buried in the closed conformation (Fig 1A). We also identified new potential elastase cleavage sites in 501Y.V2: two of the substitutions are fixed in all variants of the new South African strain (Asp80Ala and Ala701Val) and the other two are not (Asp215Gly and Arg246Ile; Fig 2B). Human neutrophil elastase prefers valine over alanine [23, 24], explaining the potential gain of sensitivity to elastase due to Ala701Val substitution in B.1.351. Note that although proteolysis-induced activation was first mapped to the S1/S2 boundary in the spike protein [27], protease-induced cuts at other sites may increase the propensity of the spike protein to expose the RBD.

## The new variants may decrease intramolecular interactions in the spike protein

We next examined a complementary possible mechanism for the new mutations to compete for the older viral variants, namely substitutions that may decrease intramolecular interactions in the spike protein, thus rendering it more likely to be in the open/active conformation. The open conformation in this trimeric protein exposes the RBD in each of the monomers, thus enabling the virus to bind to hACE2 and enter the body. Indeed, previous work suggested that the Asp614Gly substitution is more prone to be in the open conformation [28].

Using MOE, we found that Asn501Tyr substitution in both 501Y.V1 in 501Y.V2 may increase the propensity of the spike protein to be in the 'open' or active conformation (Fig 3A). The backbones of Asn501 residues in the three monomers are about 14 Å from each other in the closed trimeric structure. Therefore, a substitution with a bulky tyrosine residue (~7 Å,

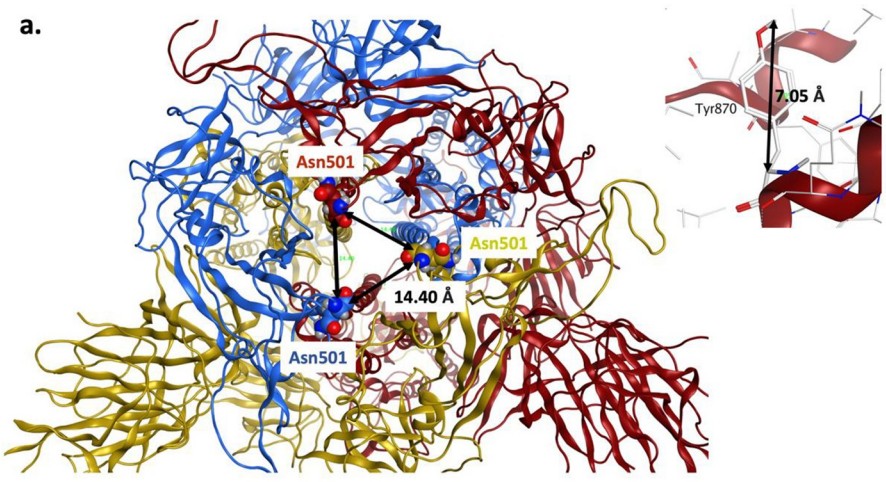

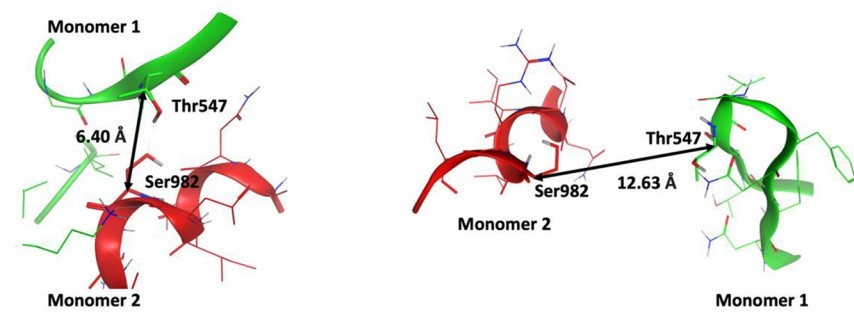

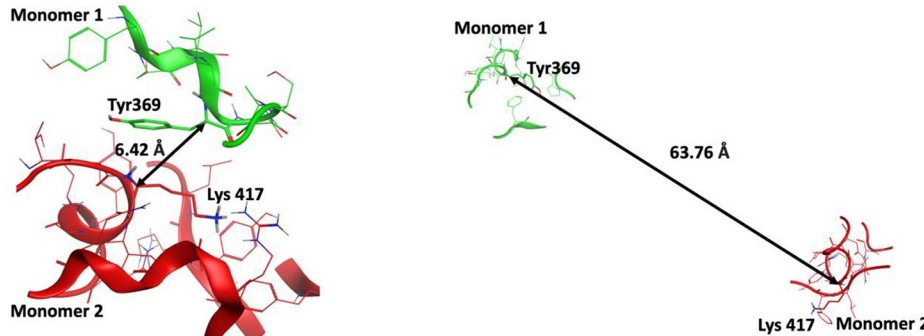

**Fig 3. Potential destabilizing effects of the point mutations on the closed conformation of the spike protein trimer. (a)** Distance between the backbones of Asn501 in closed conformation is shown (left) and typical length of Tyr side chain is shown (right). Asn501Tyr mutation might introduce steric clashes and destabilize the closed conformation. **(b)** Ser982 (monomer 2)-Thr547 (monomer 1) interaction in closed (left) and open (right) conformation. Ser982Ala results in loss of this inter-monomeric interaction. **(c)** Lys417 (monomer 2)-Tyr369 (monomer 1) interaction in closed (left) and open (right) conformation. Lys417Asn mutation may reduce this inter-monomeric interaction.

each) may introduce clashes and cause a loosening of the contact sites between the monomers in the closed conformation (Fig 3A). Note, however, MOE calculations suggest that 501Tyr substitution does not significantly destabilize the closed conformation (ΔStability 0.4369 kcal/mol) as compared with the index Wuhan 501Asn variant.

Another intramolecular interaction in 501Y.V1 that can affect transition from closed to open conformer is mediated by serine 982. Ser982 in one monomer forms a hydrogen bond with a Thr547 in an adjacent monomer in the closed conformation (Fig 3B). The substitution serine to alanine in 501Y.V1 abolishes this interaction with Thr547 in the neighboring spike monomer, thus likely favoring the open conformation.

In addition to the Asn501Tyr, the South African variant, 501Y.V2, has another amino acid substitution that may favor open conformation of the spike protein: Lys417Asn (Table 2). In the closed conformation, Lys417 present in one monomer makes a proton-π interaction with Tyr369 present in the adjacent monomer (Fig 3C). Asparagine substitution at this position may disrupt this interaction with Tyr369, potentially favoring the transition to open conformation.

Note that Thr716Ile, Ser982Ala, and Asp1118His are also conserved in viruses isolated from bat, civet and pangolin, suggesting that these amino acid substitutions may be important for some function of the spike protein (Fig 4). Similarly, Asp80Ala and Ala701Val amino acid that are mutated in 501Y.V2 are conserved in these other species (Fig 4). Whether these mutations are associated with more efficient transmissibility in various species remain to be determined.

## SARS-CoV-2 variants in the RBD

The Ala501Tyr mutation common to both 501Y.V1 and 501Y.V2 variants and Glu484Lys and Lys417Asn of the 501Y.V2 variant are among the amino acids shown to interact with hACE2 [29]. It is therefore possible that these mutations increase the binding of the active conformer of the spike protein to hACE2 or to other associated receptors to mediate more effective transmissibility. Although the affinity calculations in MOE suggest that these mutations have minimal negative impact on the affinity of RBD to hACE2 (ΔAffinity; Table 3), recent yeast display assay suggests that co-occurrence of these mutations increases the affinity of RBD to hACE2 by 2 and 10 folds for 501Y.V1 and 501Y.V2, respectively [30]. The RBD of the B.1.1.7 variant was shown to have higher affinity for hACE2 along with a lower dissociation rate constant [30]. These findings were further corroborated by the study of Khavari et al. where they used microscale thermophoresis to show that 501Y.V1 and 501Y.V2 variants bind to hACE2 with two- and five-fold higher affinity, respectively [31]. However, we suggest that mutations in RBD may also affect the closed conformation's stability, thus increasing the probability of binding to hACE2; the monomers wrap around one another through RBD in the closed trimeric structure and mutations in RBD that abolish these critical inter-monomeric interactions could favor the transition to open conformation. Therefore, the effect of these mutations, alone or in combination, on RBD-hACE2 affinity should be tested using intact viral particles.

## Discussion

### Mutations and the impact on SARS-CoV-2 neutralizing antibodies

The emergence of new virus variants triggers the concern that previous exposure to the virus, antibody therapeutics and current vaccines against SARS-CoV-2 may not be effective. This public health concern needs to be addressed as soon as possible. Many neutralizing antibodies were mapped to the RBD, and four major classes of neutralizing sites have been identified in this domain [32, 33]. As one of the mutations in SARS-CoV-2 501Y.V1 and three mutations in

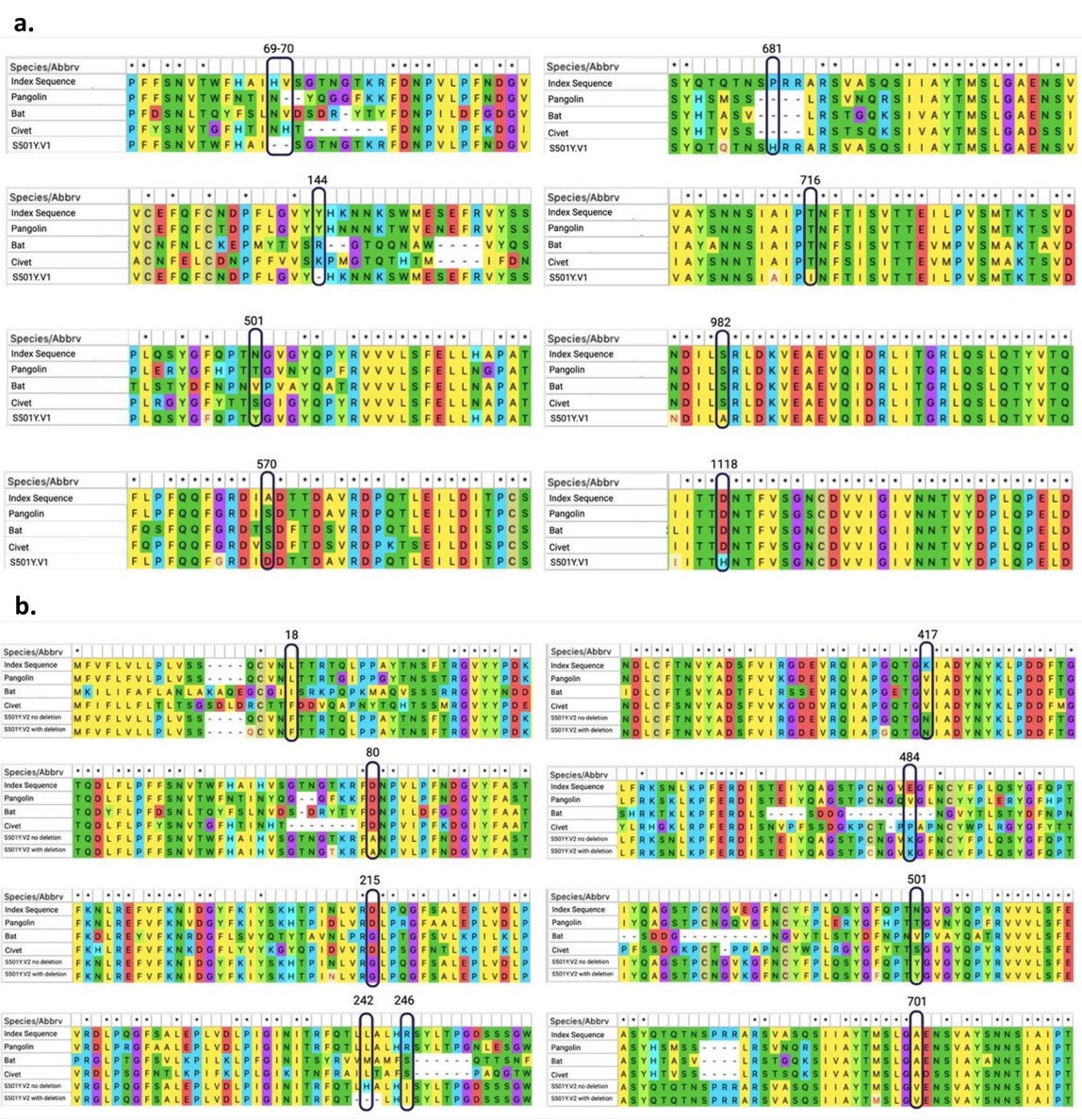

**Fig 4. Sequence alignment of the spike protein of the SARS-CoV-2 variants. (a)** Conservation of mutations found in 501Y.V1 in comparison to the Wuhan index human, bat, civet and pangolin spike protein sequences. **(b)** Conservation of mutations found in 501Y. V2 in comparison to the index Wuhan human, bat, civet and pangolin spike protein sequences. Sequence alignments generated using MEGA X software.

**Table 3. ΔAffinity calculations for RBD variants of spike protein in SARS-CoV-2 501Y.V1 and SARS-CoV-2 501Y.V2.**

| Mutation | ΔAffinity (kCal/mol) |
|---|---|
| Lys417Asn | 1.67 |
| Glu484Lys | 0.61 |
| Asn501Tyr | 2.29 |

ΔAffinity calculations for RBD variants in spike protein.

501Y.V2 are in the RBD, these mutations may decrease the neutralizing activities of antibodies [14, 34]. However, neutralization studies refute this idea; both sera of convalescent patients and immunized subjects appear effective [35–37]. Notably, mutation Glu484Lys in the 501Y. V2 lineage reduces serum binding and neutralization in circulating SARS-CoV-2 isolates [38]. Furthermore, although point mutations in the spike protein may affect the therapeutic benefit of one or a combination of two monoclonal antibodies [39], the benefit of polyclonal antibodies, such as those generated in infected individuals or when using either passive or active vaccines is less likely to be negatively impacted.

## The increased transmissibility of the new variants may expose an Achilles Heel

Our analysis indicates that many of these mutations may also increase the exposure of the RBD of the spike protein. We therefore suggest that neutralization by antibodies may increase–more antigenic determinants per each viral particle may be exposed for the antibodies to bind to. Indeed, sera from hamsters infected with Asp614Gly virus exhibit slightly higher neutralization titers against the new dominant Gly614 variant than against the index variant Asp614 [28]. Furthermore, each viral particle has as many as 80 spike proteins on each virus [40], only some of which need to bind hACE2, to induce viral entry into cells. If more spike proteins are activated and expose the antigenic RBD domain, neutralizing antibodies have an increased probability to bind to viral particles and agglutinate several particles together. The resulting agglutinated particles are less likely to have a successful fusion with the cell membrane and infection may be aborted. Increased infection efficiency due to increased propensity of the spike protein activation may also increase the vulnerability of the usually hidden spike protein core to the neutralizing antibodies.

## Nasal neutrophil elastase and viral infections–a therapeutic opportunity?

Proteomic analyses identified five proteins with increased expression in nasopharyngeal swab of subjects with high polymerase chain reaction titer for SARS-CoV-2 [21]. Among these five proteins is neutrophil elastase, whose levels increased 3-fold in SARS-CoV-2 infected *vs*. non-infected individuals. Furthermore, there is a 10-fold increase in neutrophil number in swabs from infected individuals. Elastase was also found to be critical for accelerating viral entry as well as inducing hypertension, thrombosis and vasculitis [21, 41]. As there are several approved elastase inhibitors, including Sivelestat (which is approved for acute respiratory syndrome), Alvestat (α1-antitrypsin) and Roseltide, the use of these inhibitors, perhaps as intranasal spray or drops, should be considered. Neutrophil elastase appears to cleave near the RRAR (amino acids Arg682 to Arg685) of the furin cleavage site [42]. The variant amino acids that we have discussed here are likely too far to affect the affinity of neutrophil elastase for RRAR directly, but it is possible that those hydrophobic amino acids in the variants can increase the propensity for neutrophil elastase to cleave this region of spike protein [23, 24]. In addition to neutrophil elastase, neutrophils also secrete proteinase 3, cathepsin G and NSP-4. Proteinase 3 has a similar primary sequence specificity to neutrophil elastase [23] and therefore will be equally affected by mutations. It is less likely that cathepsin G will have increased S1 proteolysis, since this protease prefers aromatic or positively charged amino acids in P1 [43] and the mutations did not generate such sites. Similarly, NSP4 has a strong preference for arginine at P1 and therefore could only augment furin cleavage sites [44] provided that they were not mutated in the given variant.

## Limitations to our analysis

Our study uses computational tools to examine the potential impact of the mutations found in SARS-CoV-2 501Y.V1 and 501Y.V2 on viral infectivity but has several limitations. We focused only on two aspects–sensitivity to proteases and increased tendency of the spike protein to be in the open conformation. Other mechanisms may increase viral transmissibility and were not examined here. For example, if proteolytic events contribute to viral inactivation at the mucosa, a decline in proteolysis susceptibility, especially to proteases present in the nose, may also contribute to increased infectivity. We also did not examine the possibility that the increased transmissibility of the new variants may be due to better survival of the virus in respiratory droplets, the potential increased affinity of the viral spike protein for receptors other than hACE2, nor the impact of mutations in other viral genes. Finally, our analysis tested mainly the potential impact of one mutation at a time. As our analysis is *in silico*, experiments to test the hypotheses raised by this study should be conducted next, to determine if these SARS-CoV-2 variants with more efficient and rapid transmission capabilities are also more sensitive to existing therapies and suggest new therapeutic targets to slow down and arrest the COVID-19 pandemic.

# Materials and methods

## Proteolysis site prediction using ExPASy

The amino acid sequence of spike protein from the reference EPI_ISL_402124 (Wuhan) sequence was uploaded to ExPASy [26] (https://web.expasy.org/peptide_cutter/) and neutrophil elastase was selected as the protease. ExPASy uses human and mouse neutrophil elastase to make cleavage site predictions. After receiving ExPASy results, we focused on human neutrophil elastase preferences to explain the potential gain in elastase cleavage sites. We examined also potential proteolysis site for the seven other proteases suggested to cleave SARS-CoV-2 spike proteins [18]. No changes in cleavage sites for both 501Y.V1 and 501Y.V2 were observed for furin, Cathepsin L/B, and PC1. However, a loss of a cleavage site was observed for matriptase in the 501Y.V1, due to alanine of serine 982 mutation. For TMPRSS2 and trypsin, which cleave at single arginine or lysine residues, a proteolysis site was gained at position 484, and a loss of proteolysis site was observed at both positions 246 and 417 in the 501Y.V2.

## Protein structures

Molecular Operating Environment (MOE) software (version 2019.01) [15] was used to generate representative structural models of the open and closed conformations of SARS-CoV-2 variant spike proteins. Protein Data Bank (PDB) ID: 7A98 [29] was used to model open conformation whereas PDB ID: 6ZB5 [45] was used for closed conformation.

## Residue scan and affinity calculation

PDB ID: 7A98 was prepared using the QuickPrep functionality at the default settings, to optimize the H-bond network and perform energy minimization on the system in MOE. The ΔAffinity calculations were performed using 7A98.A (S-protein monomer) and 7A98.D (ACE-2) chains with 7A98.D defined as ligand. Residues Lys417, Glu484 and Asn501 in S-protein (7A98.A) were selected and residue scan application was performed. The difference in affinity (ΔAffinity (kcal/mol)) between mutated residue and the wild-type were calculated as per MOE's definition.

### Residue scan and stability calculation

QuickPrep functionality was used as above to prepare PDB ID: 6ZB5 in MOE. 6ZB5.A, 6ZB5.B and 6ZB5.C chains were used to perform the calculations. Asn501 residue in S-protein in all three chains were selected and residue scan application was performed for the observed variants Ser, Tyr, Thr and Arg that are tolerated at this position with site-limit set to 3. The change in stability (ΔStability; kcal/mol) between the variants and the reference wild-type sequence were calculated as per MOE's definition.

### Calculating predicted effect of variants in PROVEAN and SIFT

The amino acid sequence of spike protein from the index EPI_ISL_402124 (Wuhan) sequence was uploaded to PROVEAN [46] (http://provean.jcvi.org/index.php) and SIFT [47, 48] (https://sift.bii.a-star.edu.sg). Every variant observed in 501Y.V1 (UK variant) and 501Y.V2 (South Africa variant) was also uploaded to compare against the reference sequence. Each variant was either predicted to be 'deleterious' or 'neutral' in PROVEAN or 'deleterious' or 'benign' in SIFT.

### Multiple sequence alignment

MEGA X software [49] and the ClustalW alignment function was used to align the amino acid sequences of the human index, bat, civet, pangolin, 501Y.V1 and 501Y.V2 SARS-CoV-2 spike proteins.

## Supporting information

**S1 Table. Stability calculations for residues in amino acid position 501 of spike protein using PDB ID: 6ZB5.**
(XLSX)

**S2 Table. Affinity calculations for variants of concern using PDB ID: 7A98.**
(XLSX)

## Acknowledgments

**General**: We are grateful to the many investigators throughout the world that continue to provide SARS-CoV-2 sequences to public databases.

## Author Contributions

**Conceptualization:** Daria Mochly-Rosen.

**Formal analysis:** Suman Pokhrel, Benjamin R. Kraemer, Lucia Lee.

**Investigation:** Suman Pokhrel, Benjamin R. Kraemer, Lucia Lee.

**Supervision:** Daria Mochly-Rosen.

**Visualization:** Suman Pokhrel, Benjamin R. Kraemer, Lucia Lee, Kate Samardzic.

**Writing – original draft:** Suman Pokhrel, Benjamin R. Kraemer, Lucia Lee.

**Writing – review & editing:** Kate Samardzic.

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
