## [Decision Letter · Decision Letter 0]

7 Apr 2021

PONE-D-21-05020

Increased elastase sensitivity and decreased intramolecular interactions in the more transmissible SARS-CoV-2 variants’ spike protein

PLOS ONE

Dear Dr. Mochly-Rosen,

Thank you for submitting your manuscript to PLOS ONE. After careful consideration, we feel that it has merit but does not fully meet PLOS ONE’s publication criteria as it currently stands. Therefore, we invite you to submit a revised version of the manuscript that addresses the points raised during the review process.

In your revised version please address as fully as possible the pertinent comments of the two reviewers.

We look forward to receiving your revised manuscript.

Kind regards,

Israel Silman

Academic Editor

PLOS ONE

Journal Requirements:

2. Please include your table as part of your main manuscript and remove the individual file.

Please note that supplementary tables should be uploaded as separate "supporting information" files.

Reviewers' comments:

Reviewer's Responses to Questions

**Comments to the Author**

1. Is the manuscript technically sound, and do the data support the conclusions?

Reviewer #1: No

Reviewer #2: Partly

2. Has the statistical analysis been performed appropriately and rigorously? 

Reviewer #1: No

Reviewer #2: N/A

3. Have the authors made all data underlying the findings in their manuscript fully available?

Reviewer #1: Yes

Reviewer #2: Yes

4. Is the manuscript presented in an intelligible fashion and written in standard English?

Reviewer #1: Yes

Reviewer #2: Yes

5. Review Comments to the Author

Reviewer #1: The authors used multiple computational methods like MOE analysis, ExPASy, and sequence alignment to evaluate the influence of 501Y.V1 and 501Y.V2 mutants of S protein on SARS-Cov-2 transmissibility. The underlying mechanism, as proposed by the authors, is that the mutations impair the intramolecular interactions of S protein and thus promote the transition from its closed conformation to the opened one, increasing the probability of binding to ACE2. However, such a hypothesis was made simply on the interaction analysis of the static structure of S protein. The structural dynamics, which might be achieved by other experimental and/or computational methods, e.g., molecular dynamics simulation, was not presented, giving no solid evidence to conduct the conclusions. Thus, I don’t think the article in the present form is eligible for publication as a research article.

Reviewer #2: The authors, Pokhrel et al., describe a decreased intramolecular interaction of the homotrimer within the spike protein of the more transmissible SARS-CoV-2 lineages in silico. Furthermore, the authors found additional neutrophil elastase (NE) cleavage sites, which are introduced in the novel SARS-CoV-2 lineages. Although the manuscript lack of experiments, the findings are of interest. The following concerns should be addressed.

Major concerns:

1. In general, the abstract is missing the findings of neutrophil elastase (NE).

2. Small aliphatic residues, in particular valine and alanine, and to a lesser extent threonine and isoleucine, are preferred by NE at P1 Kasperkiewicz et al., 2014 (10.1073/pnas.1318548111) and Schilling and Overall, 2008 (10.1038/nbt1408). These manuscripts should be cited and comparted with obtained data from the prediction tool of ExPASy. Notably, the mutation of B1.1.7 lineage shows an amino acid exchange of threonine (T) to isoleucine (I) and the B.1.351 lineage, A701V. Why would isoleucine induce a new cleave site compared to threonine or alanine to valine when both are accepted for NE in P1 position?

3. The description of the source of elastase (human or porcine; pancreas or neutrophil derived elastase) for prediction is missing, for instance human elastase shows a slightly different preference for amino acids.

Minor concerns:

1. Why 2 titles?

2. One letter amino acid code instead of three letter code should be used constantly. Otherwise it is confusing. For instance, table 1 both are used.

3. Table 1c: The manuscript regarding affinity of S protein (B.1.1.7 vs. B.1.531) to ACE2 need to be mentioned/discussed https://doi.org/10.1101/2021.02.22.432359. The authors performed “microscale thermophoresis” for binding studies. Or Collier et al., 2021 (https://www.citiid.cam.ac.uk/wp-content/uploads/2021/02/POST-SUBMISSION_vaccine-DCv2-2.pdf)

4. Page 2, introduction, 1. Paragraph: The S protein has 1273 aa, 3,540 substitutions found (bioRxiv. 2021:2021.01.04.425340), how can it be that the S protein is 4 times mutated as mentioned in the text?

5. Page 2, introduction, 2. Paragraph: B.1.1.7 variant transmissible 40-70%, seems high. New publication should be cited. To my knowledge it is rather around 35%.

6. Page 2, results, 3. Paragraph: Activated neutrophils, in addition to NE, these cells also secret proteinase 3, cathepsin G, NSP-4.

7. Figure 2. UK variant or B.1.1.7 and South African variant or B.1.351 should be written in the figure, above a) and b). Additionally, arrows to make it clear at which position the NE cleavage occurs.

8. Figure 4. Needs a higher resolution, in this form figure 4 is not readable.

9. NE and S protein, Mustafa et al. 2021 (doi.org/10.1021/acsomega.1c00363) should be discussed.

6. PLOS authors have the option to publish the peer review history of their article (what does this mean?). If published, this will include your full peer review and any attached files.

Reviewer #1: No

Reviewer #2: No

---

## [Author Response · Author response to Decision Letter 0]

13 Apr 2021

Review Comments

Reviewer #1: The authors used multiple computational methods like MOE analysis, ExPASy, and sequence alignment to evaluate the influence of 501Y.V1 and 501Y.V2 mutants of S protein on SARS-Cov-2 transmissibility. The underlying mechanism, as proposed by the authors, is that the mutations impair the intramolecular interactions of S protein and thus promote the transition from its closed conformation to the opened one, increasing the probability of binding to ACE2. However, such a hypothesis was made simply on the interaction analysis of the static structure of S protein. The structural dynamics, which might be achieved by other experimental and/or computational methods, e.g., molecular dynamics simulation, was not presented, giving no solid evidence to conduct the conclusions. Thus, I don’t think the article in the present form is eligible for publication as a research article.

Reply: In order to address this comment, we modified the title of the article to clearly indicate that this is – an in silico analysis, reflecting also the opinion of reviewer #2. 

Reviewer #2: The authors, Pokhrel et al., describe a decreased intramolecular interaction of the homotrimer within the spike protein of the more transmissible SARS-CoV-2 lineages in silico. Furthermore, the authors found additional neutrophil elastase (NE) cleavage sites, which are introduced in the novel SARS-CoV-2 lineages. Although the manuscript lack of experiments, the findings are of interest. The following concerns should be addressed.

Major concerns:

1. In general, the abstract is missing the findings of neutrophil elastase (NE).

Reply: This is now added (relates also to comment 2).

2. Small aliphatic residues, in particular valine and alanine, and to a lesser extent threonine and isoleucine, are preferred by NE at P1 Kasperkiewicz et al., 2014 (10.1073/pnas.1318548111) and Schilling and Overall, 2008 (10.1038/nbt1408). These manuscripts should be cited and comparted with obtained data from the prediction tool of ExPASy. 

Reply: We thank the reviewer, and these references are now added. 

Notably, the mutation of B1.1.7 lineage shows an amino acid exchange of threonine (T) to isoleucine (I) and the B.1.351 lineage, A701V. Why would isoleucine induce a new cleave site compared to threonine or alanine to valine when both are accepted for NE in P1 position?

Reply: We now cite reports that explain the above and have added this to the manuscript:

a. Fu et al. (2018) and Schilling & Overall (2008) demonstrate that I in the P1 site and not the T is preferred by elastase – which explains B1.1.7 T716I analysis. The publications provided by the reviewer indicate that NE can also cleave when T is in the P1 site. Therefore, preference of T or I at the proteolysis site may also depend on what other residues are present around it. 

b. Human neutrophil elastase seems to prefer V at P1 position over A, thus explaining the potential gain in B.1.351 A701V (Fu et al. 2018 and Schilling & Overall 2008 in the manuscript). 

3. The description of the source of elastase (human or porcine; pancreas or neutrophil derived elastase) for prediction is missing, for instance human elastase shows a slightly different preference for amino acids.

Reply: We agree that this is important to highlight; ExPASy utilizes both human and murine neutrophil elastase to make predictions, and both enzymes share similar catalytic activity (Wiesner et al. (2005): https://doi.org/10.1016/j.febslet.2005.08.056); we focused on the specificity of human elastase (now indicated). 

Minor concerns:

1. Why 2 titles? 

Reply: We removed the subtitle and added, ‘501Y.V1 and 501Y.V2’ and ‘an in silico analysis’ to the title (the latter per the comment of reviewer #1.)

2. One letter amino acid code instead of three letter code should be used constantly. Otherwise it is confusing. For instance, table 1 both are used.

Reply: Now corrected all to be consistent to three letter abbreviations, the more commonly used abbreviations. 

3. Table 1c: The manuscript regarding affinity of S protein (B.1.1.7 vs. B.1.531) to ACE2 need to be mentioned/discussed https://doi.org/10.1101/2021.02.22.432359. The authors performed “microscale thermophoresis” for binding studies. Or Collier et al., 2021 (https://www.citiid.cam.ac.uk/wp-content/uploads/2021/02/POST-SUBMISSION_vaccine-DCv2-2.pdf)

Reply: We have added discussion of this point in the manuscript [refs 31, 32]

4. Page 2, introduction, 1. Paragraph: The S protein has 1273 aa, 3,540 substitutions found (bioRxiv. 2021:2021.01.04.425340), how can it be that the S protein is 4 times mutated as mentioned in the text?

Reply: Corrected now to ‘almost 3 times per position’.

5. Page 2, introduction, 2. Paragraph: B.1.1.7 variant transmissible 40-70%, seems high. New publication should be cited. To my knowledge it is rather around 35%.

Reply: An increased fitness (transmissibility) of the variants has been reported in various reports over that of the Wuhan variant and, more recently, over whatever variants that was circulating at the time (e.g. D614G variant). This may explain the difference in the numbers that we cited vs. those that the reviewer cites. We now explain this point and provide a recent reference that addresses it (van Dorp et al. (2021): doi: 10.1101/2021.03.29.21254233).

6. Page 2, results, 3. Paragraph: Activated neutrophils, in addition to NE, these cells also secret proteinase 3, cathepsin G, NSP-4.

Reply: We agree with the reviewer and have added the following clarification in the discussion: ‘In addition to neutrophil elastase, neutrophils also secrete proteinase 3, cathepsin G and NSP-4. Proteinase 3 has a similar primary sequence specificity to neutrophil elastase [23]. It is less likely that cathepsin G participates in S1 proteolysis, since this protease prefers aromatic or positively charged amino acids in P1 [41] and the mutations did not generate such sites. Similarly, NSP4 has a strong preference for arginine at P1 and therefore could augment furin cleavage sites [42] provided that they were not mutated in the given variant.’ (References are numbered as in the manuscript).

7. Figure 2. UK variant or B.1.1.7 and South African variant or B.1.351 should be written in the figure, above a) and b). Additionally, arrows to make it clear at which position the NE cleavage occurs.

Reply: The figure is now modified per the reviewer’s suggestion. You did not add arrows

8. Figure 4. Needs a higher resolution, in this form figure 4 is not readable.

Reply: Now provided. 

9. NE and S protein, Mustafa et al. 2021 (doi.org/10.1021/acsomega.1c00363) should be discussed.

Reply: This study is now discussed. We have added the following text: ‘Neutrophil elastase appears to cleave near the RRAR (amino acids Arg682 to Arg685) of the furin cleavage site [42].’. 

6. PLOS authors have the option to publish the peer review history of their article (what does this mean?). If published, this will include your full peer review and any attached files.

Reply: We agree to publish the history of our article.

---

## [Decision Letter · Decision Letter 1]

27 Apr 2021

Increased elastase sensitivity and decreased intramolecular interactions in the more transmissible 501Y.V1 and 501Y.V2 SARS-CoV-2 variants’ spike protein – an in silico analysis

PONE-D-21-05020R1

Dear Dr. Mochly-Rosen,

We’re pleased to inform you that your manuscript has been judged scientifically suitable for publication and will be formally accepted for publication once it meets all outstanding technical requirements.

Kind regards,

Israel Silman

Academic Editor

PLOS ONE

Additional Editor Comments (optional):

Reviewers' comments:

Reviewer's Responses to Questions

**Comments to the Author**

1. If the authors have adequately addressed your comments raised in a previous round of review and you feel that this manuscript is now acceptable for publication, you may indicate that here to bypass the “Comments to the Author” section, enter your conflict of interest statement in the “Confidential to Editor” section, and submit your "Accept" recommendation.

Reviewer #1: All comments have been addressed

Reviewer #2: All comments have been addressed

2. Is the manuscript technically sound, and do the data support the conclusions?

Reviewer #1: Yes

Reviewer #2: Yes

3. Has the statistical analysis been performed appropriately and rigorously? 

Reviewer #1: Yes

Reviewer #2: N/A

4. Have the authors made all data underlying the findings in their manuscript fully available?

Reviewer #1: Yes

Reviewer #2: Yes

5. Is the manuscript presented in an intelligible fashion and written in standard English?

Reviewer #1: Yes

Reviewer #2: Yes

6. Review Comments to the Author

Reviewer #1: (No Response)

Reviewer #2: To the authors,

The authors addressed all concerns made by the Reviewer.

In addition, the title was changed "...an in silico analysis" in order to indicate that the analysis was done in silico.

7. PLOS authors have the option to publish the peer review history of their article (what does this mean?). If published, this will include your full peer review and any attached files.

Reviewer #1: No

Reviewer #2: No

---

## [Editor Report · Acceptance letter]

3 May 2021

PONE-D-21-05020R1 

Increased elastase sensitivity and decreased intramolecular interactions in the more transmissible 501Y.V1 and 501Y.V2 SARS-CoV-2 variants’ spike protein – an *in silico* analysis 

Dear Dr. Mochly-Rosen:

I'm pleased to inform you that your manuscript has been deemed suitable for publication in PLOS ONE. Congratulations! Your manuscript is now with our production department. 

Kind regards, 

on behalf of

Prof. Israel Silman 

Academic Editor

PLOS ONE